# Facilitators and barriers for clinical implementation of a 30-minute point-of-care test for Neisseria gonorrhoeae and Chlamydia trachomatis into clinical care: A qualitative study within sexual health services in England

Agata Pacho[1], Emma M. Harding-Esch[1,2¤a], Emma G. Heming De-Allie[1], Laura Phillips[1], Martina Furegato[1], S. Tariq Sadiq[1], Sebastian S. Fuller[1¤b]*

1 Institute for Infection and Immunity, Applied Diagnostic Research and Evaluation Unit, St George's University of London, London, United Kingdom, 2 HIV and STI Department, National Infection Service, Public Health England, London, United Kingdom

¤a Current address: Clinical Research Department, Faculty of Infectious and Tropical Diseases, London School of Hygiene & Tropical Medicine, London, United Kingdom
¤b Current address: Health Systems Collaborative, Nuffield Department of Medicine, University of Oxford, Headington, Oxford, United Kingdom
* sebastian.fuller@ndm.ox.ac.uk

## Abstract

Point-of-care tests (POCTs) to diagnose sexually transmitted infections (STIs) have potential to positively impact patient management and patient perceptions of clinical services. Yet there remains a disconnect between development of new technologies and their implementation into clinical care. With the advent of new STI POCTs arriving to the global market, guidance for their successful adoption and implementation into clinical services is urgently needed. We conducted qualitative in-depth interviews with professionals prior to and post-implementation of a *Chlamydia trachomatis/Neisseria gonorrhoeae* POCT into clinical services in England to define key stakeholder roles and explore the process of POCT integration. Participants self-identified themselves as key stakeholders in the STI POCT adoption and/or implementation processes. Data consisted of interview transcripts, which were analysed thematically using NVIVO 11. Six sexual health services were included in the study; three of which have implemented POCTs. We conducted 40 total interviews: 31 prior to POCT implementation and 9 follow-up post-implementation. Post-implementation data showed that implementation plans required little or no change during service evaluation. Lead clinicians and managers self-identified as key stakeholders for the decision to purchase, while nurses self-identified as "change champions" for implementation. Many identified senior clinical staff as those most likely to introduce and drive change. However, participants stressed the importance of engaging all clinical staff in implementation. While the accuracy of the POCT, its positive impact on patient management and the ease of its integration within existing pathways were considered essential, costs of purchasing and utilising the technology were identified as central to the decision to purchase. Our study shows that key decision-makers for adoption and implementation require STI POCTs to have

**Data Availability Statement:** The datasets generated and analysed during the current study are available in the Figshare repository, doi: 10.24376/rd.sgul.11830764.

**Funding:** This project was funded by Innovate UK Small Business Research Initiative (SBRI) grant "Stratified medicine: connecting the UK infrastructure. Phase 2" to binx health, ref: no. 90174-463338 (awarded to STS, SSF, EMHE; https://www.gov.uk/government/collections/sbri-the-small-business-research-initiative). The specific roles of these authors are articulated in the 'author contributions' section. This report is work commissioned by Innovate UK. The views expressed in this publication are those of the authors and not necessarily those of Innovate UK. The funders had no role in study design, data collection and analysis, decision to publish, or preparation of the manuscript.

**Competing interests:** The authors have read the journal's policy and have the following competing interests to declare: At the time this research was being conducted, all authors were employed by the Applied Diagnostic Research and Evaluation Unit (ADREU) at St George's University of London; ADREU has received funding from Abbott (https://www.abbott.com/), binx health (https://mybinxhealth.com/), Cepheid (https://www.cepheid.com/), SpeedDx (https://plexpcr.com/), Mologic (https://mologic.co.uk/), Revolugen (https://revolugen.co.uk/), and Sekisui (https://sekisuidiagnostics.com/), for the research and evaluation of their diagnostics. The present study was funded by a collaborative grant (ref: no. 90174-463338; awarded to STS, SSF, EMHE) between binx health and St George's University of London. This does not alter our adherence to PLOS ONE policies on sharing data and materials. There are no patents, products in development or marketed products associated with this research to declare.

laboratory-comparable accuracy and be affordable for purchase and ongoing use. Further, successful integration of POCTs into sexual health services relies on supportive interpersonal relationships between all levels of staff.

## Introduction

Point-of-care tests (POCTs) to enable to rapid and appropriate management of sexually transmitted infections (STIs) have potential to positively impact infection control, disease progression, and patient perceptions of clinical services [1–3]. Despite the potential advantages STI POCTs may bring, these do not necessarily lead to their adoption into clinical services. Many POCTs have struggled to be purchased, implemented and integrated ("adopted") into healthcare [4, 5]. Guidance for adoption of new diagnostics into clinical care are usually premised on evidence-based medicine (EBM), which is predicated on robust clinical research determining best clinical practice [6, 7]. Within England, the National Institute for Clinical Excellence (NICE) provides clinical practice guidance based on EBM [6, 8]. However, many have argued there are significant deficits to relying on EBM alone to direct clinical practice [9]. Social and contextual forces influencing new diagnostics' adoption are poorly understood and receive relatively little formal attention compared to clinical efficacy data [10, 11]. Clinical practitioners have argued that their own clinical expertise—based on years of direct practice experience—is increasingly discounted in favour of EBM's reliance on evidence from randomised controlled trials [9]. It has been suggested that key decision-makers consider social, structural and contextual factors as well as clinical trial data in their decision to implement new technologies [6, 12].

Enquiries into facilitators and barriers to POCT adoption are particularly timely for sexual health: new rapid and POCTs for *Chlamydia trachomatis* (CT) and *Neisseria gonorrhoeae* (NG) are becoming increasingly available for clinical adoption [13]. The dual CT/NG nucleic acid amplification test (NAAT) provided by binx health is one such POCT; qualitative diagnosis of CT/NG is provided on the $io^{®}$ system within 30-minutes of sample input [13]. This POCT is currently CE-marked for use in Europe, and has received Food and Drug Administration (FDA) clearance in the USA for CT/NG detection in genital samples (female vaginal swabs and male urine; data currently under review by FDA for CLIA waiver) [13]. The *Facilitators to Adoption* study (2017–19) was developed to explore key social, structural and contextual facilitators and barriers in the process of adoption of the binx health io CT/NG Assay (the "binx POCT") into sexual health services in England.

## Methods

We conducted a qualitative study of healthcare professionals (HCPs), commissioners and managers in English sexual health services, between April 2018 and October 2019. This study was conducted concurrently with an experimental model that sought to de-risk the adoption of the binx POCT by supplying instruments and CT/NG cartridges free-of-charge to clinical services, and assisting with data analysis to enable service evaluation of the binx POCT (with the ultimate goal of clinical adoption). The experimental model is reported elsewhere [14].

Invitation to the qualitative study was based on services' participation in the experimental model. Invitation to participate in the experimental model was advertised through the British Association for Sexual Health and HIV newsletter, to our previous collaborators and by word-of-mouth. Initially, 14 services expressed interest; six of these withdrew following telephone

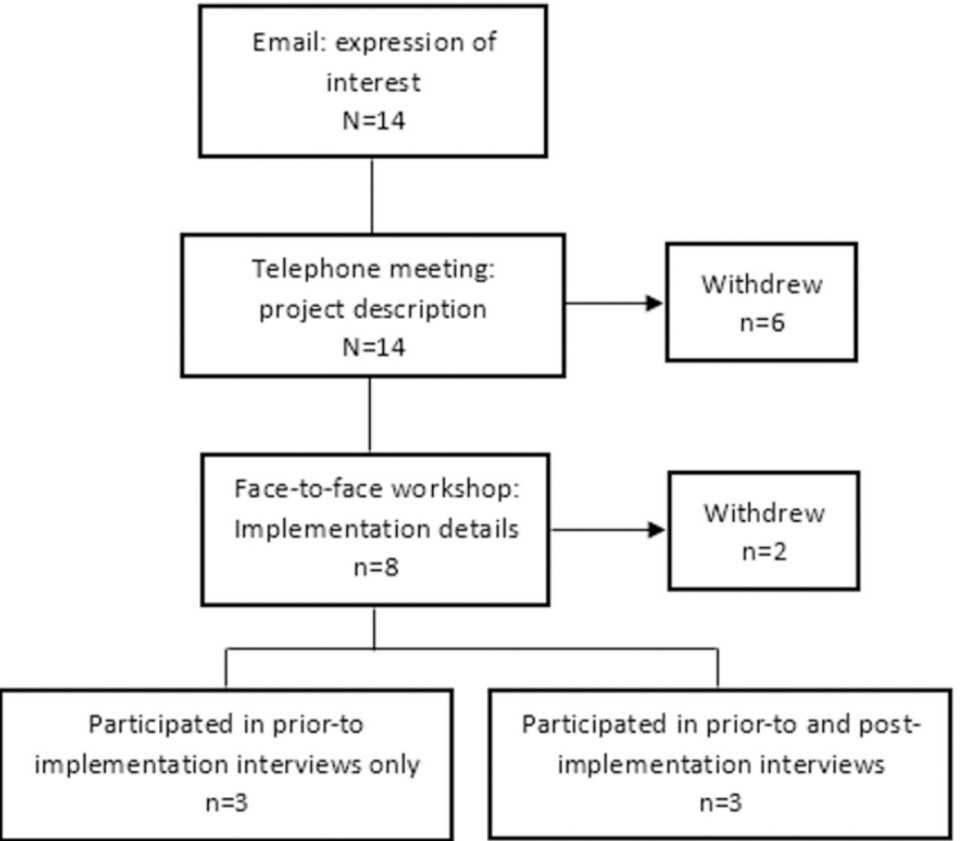

**Fig 1. Service recruitment and participation in qualitative interviews.**

calls between the service leads and the research team to discuss the premise and mechanisms of the project. Services that withdrew felt the binx POCT would not be a good fit for their service, or could not supply (human) resources to take part in the experimental model. Two additional services withdrew following participation in a workshop where the study team and self-identified key stakeholders from the services discussed the practicalities of the project, and implementation and adoption. The two services that withdrew cited: belief that the binx POCT would increase cost / not be cost-efficient for their service, and lack of space to house even a single binx POCT instrument. Six services took part in the pre-implementation interviews, but three of these ultimately declined participation in the experimental model. Reasons for withdrawal included administrative and planning difficulties, and stakeholders reporting they felt there was a poor fit between the POCT and local clinical population. The latter resulted from the device not being approved for extragenital sampling, which was seen as a key requirement for clinics with high numbers of male patients reporting sex with men. The service recruitment process is shown in Fig 1.

For confidentiality purposes, participating services are not named in this report. Of the six services participating in the pre-implementation interviews, two are located in inner London, one is located in a mid-sized city in north England and three are peri-urban services located in south-east England. Service footfall in those clinics ranges considerably, with the smallest service seeing ~500 patients per month to larger services seeing ~4000 patients a month. Of the three services that participated in the experimental model, Service 1 is a high-throughput service seeing an average of 2250 patients a month, located in inner London; Service 2 is a high

throughput service located in south-east England seeing a combined average of 3550 patients a month; and Service 3 is a lower throughput service located in south-east England and sees a combined average of 500 patients a month, with testing conducted in general practice surgeries as well as the main sexual health clinic.

Qualitative, in-depth interviews were conducted with professionals identified via the experimental model who self-identified as key in either the decision-making process for adoption of new technologies, and/or implementing new technologies, into their current service. Informed consent was obtained from all respondents; those interviewed face to face provided written consent; verbal consent was recorded prior to interviews conducted over the phone. An initial set of interviews was conducted prior to the binx POCT implementation to explore: key decision-makers; structural and social facilitators/barriers to adoption of new technologies (specifically: diagnostics, information technology) into sexual health services; their opinions of the usefulness of POCTs for CT/NG, and POCTs for multiple STI infections, to their current service (Topic Guide1; supplement 1). Follow-up interviews were planned to take place post-implementation of the binx POCT to explore any changes in rationale or plans described in initial interviews (Topic Guide2; supplement 2). In pre-and post-implementation interviews, respondents were asked to describe service evaluation (prospective or actual) implementation processes in their own words and to identify facilitators and barriers to POCT adoption.

This study (reference 224413) received approval from the National Health Service (NHS) Health Research Authority; it did not require further NHS Research Ethics Committee review.

Interviews were conducted by AP, and analysis conducted by AP and SSF; both have >10 years' experience with qualitative research. Interviews were conducted one-on-one, either face-to-face or via telephone depending on participants' preference, audio recorded and transcribed verbatim. Transcripts for both sets of interviews were then checked for accuracy against the audio recording by the interviewer and imported into NVivo 11 for analysis.

Transcripts were analysed using a thematic approach, where common themes are identified across the dataset [15]. AP led the initial analysis and selected initial codes. SSF then reviewed a random selection of transcripts, generated and assigned codes independently. All codes were agreed on by SSF and AP prior to finalising analysis. A second analysis was conducted following post-implementation interviews to check dataset comparability, which was assessed independently by AP and SSF and agreed upon. Additional codes were added to the post-implementation interviews analysis to identify and explain new concepts within the data.

Comparison of implementation plans discussed in interviews conducted pre- and post-implementation did not show significant differences. Therefore, we present our analysis here as a whole, grouped under sub-headings to differentiate considerations respondents had for post-study purchase versus those that focused primarily on product implementation. A full list of nodes is shown in Fig 2 and Fig 3. We report our findings following the Completed Standards for Reporting Qualitative Research (SRQR) (S1 Checklist).

## Results

Invitation to participate was sent to professionals identified via the experimental model who self-identified as key in either the decision-making process for adoption of new technologies and/or implementing new technologies, into their current service. Forty-eight invitations were sent to HCPs (including general practitioners who offered STI testing), commissioners and clinic managers; 31 (64.6%) participants took part in initial interviews. HCPs (clinicians, nurses, health advisors and healthcare assistants) comprised the majority of respondents (71%) (Table 1).

Professionals in clinical services that ultimately declined participation in the experimental model (3/6; 50%) were not eligible to participate in post-implementation interviews.

```
NVivo Nodes: 'Prior to implementation' interviews

Clinic
    clinic experience with service evaluation and adoption
    culture of clinic
    future plans
    interpersonal relationships
    pathways and services
Drivers*
    data
    need
Financial considerations
Interpersonal*
    attitudes towards change
    communication
Other contextual forces*
    current climate of SHS
    red tape
Personal
    collaborations
    day to day responsibilities
    experience with service evaluation and adoption and role in the
    process
    personal motivation
    role
Product
    accuracy
    cost
    design and development
    fit with existing pathways and services
Service evaluation and adoption processes
    key decision makers
        commissioners
    key to success*
    required*
Structural
    clinic's capacity
    lab's capacity and contracts
```

**Fig 2. NVivo nodes 'Prior to implementation' interviews.** Asterisks represent nodes that emerged independently of the questions asked by the interviewer (e.g. inductive nodes).

Invitations to participate were sent to the 15 previous participants in the remaining three clinical services; nine of these (60%) participated. Of the nine respondents, the majority (7/9; 78%) were HCPs (Table 2).

The interviewer, (AP), presented herself to respondents as an independent University researcher with no formal ties to the POCT company, in order to illicit both positive and negative comments about the tests. AP had prior experience of conducting research in sexual health, and as such, had a history of professional collaboration with some of the participants.

```
NVivo Nodes: 'After implementation' interview

Attitudes
Clinic facilities
Collaboration and communication*
Impact
    expected impact
    patient experiences & expectations*
Leading*
Local considerations for implementation*
Process
    implementation process
    service evaluation/local validation data
    service evaluation process
    test impact lab
    which patients
    workload changes
Role in the process
Test
    test cost
    test design
    test turn-around-time
```

**Fig 3. NVivo nodes 'After implementation' interview.** Asterisks represent nodes that emerged independently of the questions asked by the interviewer (e.g. inductive nodes).

Any familiarity with the participants had no impact on the questions asked during the interviews. There was no significant difference with respect to depth of responses, or reported feelings about use of the POCTs, between data collected from respondents who had prior history with AP, and those that did not.

## The case for purchase: Perceived POCT costs and benefits

As sensitivity and specificity of the binx POCT was expected to be comparable with existing lab-based technologies, the costs of purchasing and utilising the new tests were identified as central to the decision to purchase. Whenever financial cost was discussed, it was stressed by respondents that POCTs would have to be proven to either be cost saving or cost neutral to

**Table 1. 'Prior to implementation' interviews.**

| Clinic | Number of respondents | HCPs (%) | Lab (%) | Management (%) | Commissioners (%) |
|---|---|---|---|---|---|
| Clinic 1 | 6 | 4 (66.7%) | 0 (0%) | 2 (33.3%) | 0 (0%) |
| Clinic 2 | 5 | 4 (80%) | 0 (0%) | 1 (20%) | 0 (0%) |
| Clinic 3 | 5 | 2 (40%) | 1 (20%) | 1 (20%) | 1 (20%) |
| Clinic 4 | 6 | 5 (83.3%) | 1 (16.7%) | 0 (0%) | 0 (0%) |
| Clinic 5 | 4 | 3 (75%) | 0 (0%) | 1 (25%) | 0 (0%) |
| Clinic 6 | 5 | 4 (80%) | 0 (0%) | 1 (20%) | 0 (0%) |
| **Total** | **31 (100%)** | **22 (71%)** | **2 (6.5%)** | **6 (19.3%)** | **1 (3.2%)** |

**Table 2. 'After implementation' interviews.**

| Clinic | Number of respondents | HCPs (%) | Management (%) |
|---|---|---|---|
| Clinic 2 | 2 | 2 (100%) | 0 (0%) |
| Clinic 3 | 2 | 1 (50%) | 1 (50%) |
| Clinic 6 | 5 | 4 (80%) | 1 (20%) |
| **Total** | **9 (100%)** | **7 (78%)** | **2 (22%)** |

meet the benchmark of a successful business case for purchase. Respondents referred to insufficient funding as the main reason why services do not adopt new technologies. This health advisor told us:

'And I find that a bit unfortunate. That it's all about cost. We can say that it's about patient choice and about patient experience, and everything. But the most important thing in this current state of the [health service] is that it costs less. And that's a bit frustrating sometimes, really.'–Health Advisor1, (C6)

Similarly, one clinician told us that even if new technology had potential to improve service provision, cost may still act as 'red tape' in the purchase process:

'The biggest barrier to anything is the finances. If you're trying to introduce something that may be a better way of doing it, but if it costs more, then it's very, very hard to introduce that. So, I think that would be the main barrier.'–Clinician1, (C4)

Although restricted funding was discussed by most respondents as a barrier, some felt that the current economic climate may encourage innovation:

'It's a mix . . . the current difficult or stressing financial climate gives an opportunity for things to be reviewed also so that you don't look at doing the same thing. You're under pressure to ask yourself, is there is a better way? Is there a more cost-effective way?'–Clinician1, (C5)

Other respondents also discussed the value POCTs may add to the service that were beyond comparisons of incoming and outgoing costs. Keeping the service "modern," as well as cost-effective, was seen by some respondents as providing the potential to avoid funding cuts to the service in the future.

'Well, I think you have to keep the service modern. . . . I won't be someone who puts everything in right up front without the evidence. But I do like to be an early adopter rather than dragged to the gate as it were.–Clinician1, (C2)

### Drivers of POCT implementation

Clinical leads with positive attitudes towards new technologies were seen to foster the potential to bring innovation into their services. Respondents stressed that adoption of new technology required commitment from HCPs in leading roles in the service. These individuals were recognised as most likely to introduce new ideas and drive change:

'I think these ideas generally come from the frontline. So I would say whoever the staff member is who's come up with the idea is obviously key. I think the clinical lead, the [lead nurse] and the service manager, they're like a trio that cover all areas of the service. They're the ones that in the first instance should be deciding if [a new technology] is something we want to progress.'–General Manager1, (C6)

'I'd say the climate at the moment within the department is quite exciting and people are will-
ing to try new things. And there's an eagerness and an enthusiasm within the senior doctor
team to start to move things forward definitely. And as I say our lead nurse is very proactive
in new technologies and bringing us, wanting to lead us forward.'–Nurse/Health Advisor1,
(C4)

In interviews prior to implementation, most suggested that implementing changes to
healthcare provision was enabled by supportive relationships and interactions between all
members of staff. Respondents felt that it was crucial that junior staff be included in the pro-
cess and given an opportunity to voice their opinions of changes:

'I think when it comes to [service] evaluation, everyone is involved. It might not be all coming
together at once, but it might be various teams getting feedback from various people. . . .
when it comes to evaluation, everyone's input is valid.'–Nurse1, (C2)

## POCT suitability

The design of the binx POCT was discussed by respondents as an important feature for consid-
eration of how it would be used. For example, respondents discussed that a turn-around-time
of 30 minutes would not enable the binx POCT to be used with all patients in a busy clinical
setting. Respondents from one of the three services participating in clinical implementation
raised this an issue of concern, and a consultant from another service planned to re-design
pathways so patients were occupied with other tasks (e.g. blood tests) while waiting for POCT
results, to shorten the time patients felt they were waiting. Another idea to mitigate longer wait
for results involved introduction of a sample-first pathway with POCT patients, whereby
patients would provide sample at triage and be brought into the consultation room only when
their POCT results were ready. Several respondents felt that this test would only be suitable for
selected patient groups (e.g. symptomatic patients, or sexual contacts of those found positive
for infection). Yet this was not seen negatively: respondents explained that use of the POCT
would bring significant value to their service. In particular, one clinician discussed that the
desire for a one-size-fits-all test for STIs should not predicate use of less-than-ideal
technologies:

'You end up surrounded by people who want to get the perfect, you know, the perfect scenario
before you launch into a new project. . . And then what happens is, nothing happens . . . So,
the perfect is the enemy of the good. . . . you just have to make good. And set it up, run it,
evaluate it.'–Clinician1, (C6)

Respondents interviewed prior to implementation stressed the importance of published
accuracy of the binx POCT compared to existing laboratory tests; the decision to bring the
binx POCT into the clinic was predicated on comparable sensitivity and specificity. This topic
was raised again in post-implementation interviews, with some suggesting it would take time
to build trust in the accuracy of the binx POCT:

'Do we trust the brand-new machine that we've got or do we trust the one that we've been
using for a long time?'–Healthcare Assistant1, (C1)

This same HCP went on to explain her nuanced understanding of accuracy and what that
means in the current context of laboratory results:

'So that's tricky, but the way that I rationalised it in the end was that you're going to get false
positives either way. When we send off our swabs anyway now we can't fully trust that
they're going to be correct.'–Healthcare Assistant1, (C1)

However, the ability of the binx POCT to allow HCPs to provide results to patients during their visit was discussed by some as increasing their anxiety about the accuracy of the test:

'I think it's the emotional attachment to the result, and to the patient. Because I'm thinking, I want this to be the correct result for this patient who is sitting in front of me. Whereas if you're giving them a result over the phone, or by text, it's anonymous, it's kind of colder. It's detached.'–Nurse3, (C1)

Respondents also focused on other practicalities of implementation. In post-implementation interviews, those that worked closely with the binx POCT were more likely to point out disadvantages. For example, while a nurse processing samples described the binx POCT as 'noisy', a clinician working in the same clinic that did not spend time in the room where it was placed described them as 'completely fine' and 'pretty neat'. Further, although the binx POCT is described as a 'a small, desktop instrument' [16], a nurse implementing the device highlighted lack of space in the clinic for multiple instruments.

## Discussion

To our knowledge, this is the first study among stakeholders in clinical care who have implemented a 30-minute STI POCT in practice. The factors that supported the decision to plan for adoption of the binx health *io* CT/NG Assay into participating sexual health services included its usability within each local setting and the test being regarded by key decision-makers, such as clinical leads, as fit-for-purpose when compared with existing diagnostic technologies.

It has been argued that POCTs need to be affordable to enable their successful implementation [17]. The structure and funding of sexual health services within NHS England have undergone significant changes over the past decade, with services tendered to assure best value-for-budget [18, 19], part of a global trend to reduce the cost of health services [20]. Our data indicate that within the process of decision-making, value of a new technology was considered within the broader economic and political context affecting the health service, such as a tendering process that values services' innovation. Potential political gains for modernising the service were described as nuancing decision-making processes for purchase. In addition, although upfront and ongoing costs of POCTs were considered potentially insurmountable barriers, these continued to be measured against potential patient and public health benefits by many respondents.

Our data indicate that, among our respondents, the need for solutions for specific aspects of individual service provision was greater than the desire for a one-size-fits-all POCT for STIs. While research has shown that UK patients attending sexual health services are likely to wait in clinic for POCT results in some circumstances [21], it has also been speculated that, in other contexts, when STI POCTs become widely implemented, the number of patients seeking testing may decrease due to apprehension over immediate results delivery [22]. We recommend further study of patients' opinions of routine STI POCTs, under multiple implementation conditions, to provide better understanding of best implementation pathways for STI POCTs.

In agreement with previous suggestions in the literature [23, 24], attitudes of HCPs and managers towards innovation and technology are critical to the decision to purchase, and for successful implementation in practice. In our study, this was particularly salient if the views of lead clinicians and senior nursing staff were shared by junior HCPs. These findings are particularly valuable; it has been previously found that acceptability of POCTs among HCPs is crucial, yet there has been relatively little research done among HCPs regarding implementation [25, 26].

Previous research suggests that accuracy of a new POCT is one of the most significant factors considered by key stakeholders in the adoption process [27–29]. It has been recognised that trust in new technologies is established not only through evidence of its published accuracy but also through comparison to similar technologies used in the past [30, 31]. Sexual health services in the UK often deliver laboratory-based CT/NG results to patients via text message. With the introduction of a CT/NG POCT, HCPs are likely to provide patients with their results during their consultation. Our data show that the presence of the patient at the moment of delivering their diagnosis may increase HCPs' anxiety about the accuracy of POCT results. We recommend this be given special consideration during implementation plans.

## Limitations

These interviews were conducted with staff from clinical services that expressed interest in taking part in a service evaluation of a CT/NG POCT. While the service decision to participate in the study does not imply the enthusiasm of all respondents, it is likely that those participating in the study were more likely than those who did not participate to have interest in novel technologies, POCTs in particular, and be interested in adopting them. Consequently, there is a potential bias in our data for greater favourable opinions towards STI POCTs and other novel technologies than is generally present among this population.

Most respondents were HCPs (71% in 'prior to implementation' interviews and 78% in 'after implementation' follow-up interviews), and we only interviewed one commissioner. As a result, we were not able to consider a wide range of opinions of those responsible for commissioning NHS services and contributing to this higher-level decision-making, especially with regards to the economic and financial aspects associated with adoption of new technologies.

In addition, we were unable to observe the purchase of the POCT into routine clinical service due to delays in bringing the POCT to market. Subsequently, our post-implementation interviews were focused on discussion of implementation rather than purchase.

## Conclusions

This article reports on the first-of-its-kind investigation of the implementation of a 30-minute molecular POCT for CT/NG into practice in sexual health services. This is a timely enquiry given the increase in commercial availability of rapid and POC tests for STIs, and the relative lack of data on social and contextual forces influencing new diagnostics' adoption.

Our data report on key facilitators and barriers of POCT adoption from the perspective of HCPs, and show that nuances in decision-making around POCT adoption are likely to reflect local contexts, such as service tendering concerns and patient population needs. We also found HCP views of the POCTs are likely dependent on their role within the implementation process, underlining the need to include all levels of HCPs in clinical implementation plans. We recommend further research to explore the interaction between patient acceptability and implementation of new POCTs, to further understandings of how POCT use might improve clinical and public health outcomes in today's resource-constrained health services.

## Supporting information

**S1 Checklist. SRQR checklist.**
(PDF)

## Acknowledgments

We are grateful to all participants. We also thank the members of our advisory committee: Lucy Parker, Kate Folkard, Cath Mercer, Sue Eaton, Merle Symonds, Gary Whitlock and Chris Price.

## Author Contributions

**Conceptualization:** Emma M. Harding-Esch, S. Tariq Sadiq, Sebastian S. Fuller.

**Formal analysis:** Agata Pacho, Sebastian S. Fuller.

**Funding acquisition:** Emma M. Harding-Esch, S. Tariq Sadiq, Sebastian S. Fuller.

**Investigation:** Agata Pacho.

**Methodology:** Sebastian S. Fuller.

**Project administration:** Agata Pacho, Sebastian S. Fuller.

**Supervision:** Sebastian S. Fuller.

**Writing – original draft:** Agata Pacho, Sebastian S. Fuller.

**Writing – review & editing:** Agata Pacho, Emma M. Harding-Esch, Emma G. Heming De-Allie, Laura Phillips, Martina Furegato, S. Tariq Sadiq, Sebastian S. Fuller.

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
