## [Decision Letter · Decision Letter 0]

15 Sep 2021

PONE-D-21-14685Facilitators and barriers for clinical implementation of a 30-minute point-of-care test for Neisseria gonorrhoeae and Chlamydia trachomatis into clinical care: a qualitative study within sexual health services in EnglandPLOS ONE

Dear Dr. Fuller,

Thank you for submitting your manuscript to PLOS ONE. After careful consideration, we feel that it has merit but does not fully meet PLOS ONE’s publication criteria as it currently stands. Therefore, we invite you to submit a revised version of the manuscript that addresses the points raised during the review process.

We look forward to receiving your revised manuscript.

Kind regards,

Remco PH Peters, MD, PhD, DLSHTM

Academic Editor

PLOS ONE

Journal Requirements:

3. Thank you for stating the following in the Competing Interests/Financial Disclosure* (delete as necessary) section:

“I have read the journal's policy and the authors of this manuscript have the following competing interests: ADREU has received funding from Abbott, binx health, Cepheid, SpeedDx, Mologic, Revolugen and Sekisui, for the research and evaluation of their diagnostics. STS has been an invited technical consultant for the WHO for POCTs for STIs. SSF has recieved funding from and been an invited technical consultant for the WHO for implementation of POCTs for STIs.”

We note that one or more of the authors are employed by a commercial company: Abbott, binx health, Cepheid, SpeedDx, Mologic, Revolugen and Sekisui

Reviewers' comments:

Reviewer's Responses to Questions

**Comments to the Author**

1. Is the manuscript technically sound, and do the data support the conclusions?

Reviewer #1: Partly

Reviewer #2: Yes

Reviewer #3: Yes

2. Has the statistical analysis been performed appropriately and rigorously? 

Reviewer #1: N/A

Reviewer #2: Yes

Reviewer #3: N/A

3. Have the authors made all data underlying the findings in their manuscript fully available?

Reviewer #1: No

Reviewer #2: Yes

Reviewer #3: Yes

4. Is the manuscript presented in an intelligible fashion and written in standard English?

Reviewer #1: Yes

Reviewer #2: Yes

Reviewer #3: Yes

5. Review Comments to the Author

Reviewer #1: Review: Facilitators and barriers for clinical implementation 1 of a 30-minute point-of-care test for Neisseria gonorrhoeae and Chlamydia trachomatis into clinical care: a qualitative study within sexual health services in England.

In this study, sexual health clinics opted into using Point of Care Testing (POCT) for select STI’s in England with the objective of understanding the barriers and facilitators of uptake. Utilizing in depth interviews, health professionals provided their experience and perspective on POCT including considerations for clinic uptake and adoption. The study is novel in that it examines what would make POCT successful given the low uptake of such technologies by clinics globally. The manuscript is strong but requires some revision as outlined below:

1. There are several awkward and unclear, run-on sentences throughout that need to be edited such as:

Despite the potential advantages STI POCTs may bring, their launch into market will not necessarily lead to their adoption into clinical services; many POCTs have struggled to be purchased, implemented and integrated (“adopted”) into healthcare.

With the introduction of a CT/NG POCT, HCPs are likely to provide patients with their results during their consultation; our data show that proximity to the patient in the moment of delivering their diagnoses may increase HCP anxiety about the accuracy of POCT results; we recommend this be given special consideration during implementation plans.

2. In the Methods, Lines 123-125, the domains for the interview guide need to be clarified. Also, the use of ‘probe’ is incorrect as probing is a standard practice in interviews to ask participants to elaborate on what they said. This does not need to be reported. However, it seems that interview domains are: 1) implementation experience and 2) identifying facilitators and barriers to adoption.

3. In the Methods, how did you identify themes? Did you code? If so, please describe this coding process including the development of the codebook. The analytical methods need elaboration.

4. In the Results, Paragraphs starting on lines 235, 249 and 273 seem to be new themes. If not, then more detail is needed to show how these areas are linked to the existing theme. The POCT implementation theme may be too broad as a theme itself. The quote on line 244, needs to be better integrated into the theme for the paragraph above. Also, here, it is unclear if you are talking about clinic clients or healthcare workers. Respondents and participants are used interchangeably. Suggest replacing both with healthcare worker.

5. The conclusion summarizes the Discussion again. Suggest returning to the more global issue of POCT uptake.

Reviewer #2: The manuscript by Pacho et al. addresses several issues related to the process of adoption of point-of-care tests for chlamydia and gonorrhoea into sexual health services in England. It is an interesting and informative paper and discusses some of the real-world barriers to implementation of these technologies. Given the current and future availability of such tests, and the potential benefits that they may bring, this is a very important area of research.

Overall, the manuscript was well written, and I suggest only a handful of edits to provide more information on contextual factors and to improve clarity and consistency.

Abstract

1. Line 41-42: “We conducted 40 total interviews; 31 prior to POCT implementation; 9 post-implementation.”

This sentence could indicate that 40 different individuals were interviewed. Please make it more clear in the abstract that the 9 post-implementation interviews were with participants who had also provided pre-implementation interviews.

Background

2. Line 78-80: “Enquiries into facilitators and barriers to POCT adoption are particularly timely for sexual health: new rapid and POCTs for Chlamydia trachomatis (CT) and Neisseria gonorrhoeae (NG) are becoming more commonplace”

The authors reference the WHO diagnostic landscape for STIs report to justify this statement that new POCTs for chlamydia and gonorrhoea are “becoming more commonplace”. The term commonplace is a bit ambiguous here, and potentially implies that their adoption in clinical settings has become more frequent.

If the intent of this statement was to inform the reader that there are a number of POCTs for CT and NG currently available and more are likely in in the coming years, I suggest stating this more directly and potentially mentioning the pipeline of diagnostics specifically.

However, if the authors wished to state that the use of POCTs for CT/NG is in fact more commonplace in clinical practice, an alternative reference would be required to justify this.

Methods

3. Line 102-110:

I appreciated the description of the reasons why services that initially expressed interest did not proceed. However, three of the six services that took part in the pre-implementation interviews “ultimately declined participation in the experimental model”, but the reasons for this are not discussed. For completion, please consider providing information on the reasons behind these three services not taking part, if available.

4. Line 127-128: “Interviews were conducted by AP, and analysis conducted by AP and SSF; both have >10 years’ experience with qualitative research.”

Given that this is a purely qualitative study, there is an onus to provide information on factors that may have influenced the analysis itself. Please provide more information on the background of researchers AP and SSF. The SRQR checklist recommends providing information on “researchers’ characteristics that may influence the research, including personal attributes, qualifications/experience, relationship with participants, assumptions, and/or presuppositions; potential or actual interaction between researchers’ characteristics and the research questions, approach, methods, results, and/or transferability”.

5. Please provide a completed qualitative research checklist (SRQR or COREQ) as an appendix, and state and reference this in the methods.

6. I note that the names of the clinics have been withheld. If this is for confidentiality purposes, I suggest stating this explicitly in the methods.

Additionally, if the names are to be withheld, some background could still be provided on the clinics. For example, the cities/towns in which the six clinics are based could be stated without saying which location corresponds to which clinic. Alternatively, if they are all based in cities/large towns and if they are all based at large tertiary centres/smaller community clinics etc could be stated. This may help to provide additional context to the analysis.

7. Line 146-150

The methods state that “A full list of themes is shown in Figure 2 and Figure 3.”

The titles of figures 2 and 3 refer to both “NVivo nodes” and “inductive themes”.

If the terms “NVivo nodes” and “themes” are equivalent, I suggest the use of a single term for consistency.

Results

8. Line 153-154: “Invitation to participate was sent to professionals working in six sexual health services that showed interest in participating in the experimental model.”

The methods description of the interviewees is “professionals identified via the experimental model who self-identified as key in either the decision-making process for adoption of new technologies, and/or implementing new technologies, into their current service.”

These two descriptions in the methods and results sections are slightly incongruent. The methods section description suggests that all participants were key to adoption or implementation. However, the results section suggests that interest in the experimental model was the only pre-requisite.

I suggest re-writing one description or the other to ensure these descriptions are consistent and accurate.

Discussion

9. Line 301: “Our data indicate that a need for solutions to increase patient outcomes and overall wellbeing drove many HCPs’ desire for implementation of the binx POCT.”

I do not feel that this statement was adequately supported by the analysis presented in the results section and it seems to be the first mention of the drivers behind HCPs wanting to implement the binx POCT.

Please mention this in the results section if you wish for it to be a discussion point.

Limitations

10. The majority of respondents were healthcare professionals, and only one commissioner contributed to the pre-implementation data. I suggest commenting on how the breakdown of your respondents may have affected your results.

Additionally, did the authors feel that nine post-implementation interviewees was sufficient? Was data saturation reached?

Reviewer #3: Overall an important study related to implementation of new STI diagnostic technologies. Comments are minor

- I don't see reference 14 anywhere in the text of the manuscript, but I would suggest not using a corporate press release. Similarly, reference 16 is from the manufacturer's package insert and thre are several peer-reviewed articles available describing the io so it would be better to use one of those.

- How many io instruments were supplied to each clinic? If more than 1, is this part of the issue with space?

- Did anyone mention patients having to wait >30 minutes because the instrument was already in use when their sample was taken? Is this an issue of concern for clinicians?

- Did anyone mention results integration concerns (or describe this as an advantage of the system)?

6. PLOS authors have the option to publish the peer review history of their article (what does this mean?). If published, this will include your full peer review and any attached files.

Reviewer #1: No

Reviewer #2: No

Reviewer #3: No

---

## [Author Response · Author response to Decision Letter 0]

7 Jan 2022

Dear Reviewers, 

We are grateful to you for taking the time to assess our manuscript and for your insightful comments. We have been able to incorporate changes to reflect most of the suggestions provided. We have highlighted the changes within the manuscript. 

Here is a point-by-point response to the reviewers’ comments and concerns.

Comments from Reviewer #1

• Comment 1: There are several awkward and unclear, run-on sentences throughout that need to be edited such as:

Despite the potential advantages STI POCTs may bring, their launch into market will not necessarily lead to their adoption into clinical services; many POCTs have struggled to be purchased, implemented and integrated (“adopted”) into healthcare.

With the introduction of a CT/NG POCT, HCPs are likely to provide patients with their results during their consultation; our data show that proximity to the patient in the moment of delivering their diagnoses may increase HCP anxiety about the accuracy of POCT results; we recommend this be given special consideration during implementation plans.

Response: We have rewritten those sentences for clarity (Lines: 64-65 and 350-353)

• Comment 2: In the Methods, Lines 123-125, the domains for the interview guide need to be clarified. Also, the use of ‘probe’ is incorrect as probing is a standard practice in interviews to ask participants to elaborate on what they said. This does not need to be reported. However, it seems that interview domains are: 1) implementation experience and 2) identifying facilitators and barriers to adoption.

Response: The interview domains have been clarified (Lines 146-147)

• Comment 3: In the Methods, how did you identify themes? Did you code? If so, please describe this coding process including the development of the codebook. The analytical methods need elaboration.

Response: The process of identifying codes is now explained clearly (Lines 159-173). 

• Comment 4: In the Results, Paragraphs starting on lines 235, 249 and 273 seem to be new themes. If not, then more detail is needed to show how these areas are linked to the existing theme. The POCT implementation theme may be too broad as a theme itself. The quote on line 244, needs to be better integrated into the theme for the paragraph above. Also, here, it is unclear if you are talking about clinic clients or healthcare workers. Respondents and participants are used interchangeably. Suggest replacing both with healthcare worker.

Response: Mentioned paragraphs are now under new subsections (Lines 247 and 274). The quote has been integrated in the text (Line 287-294). As not all of our respondents were healthcare workers, we have decided to use the word ‘respondents’. 

• Comment 5: The conclusion summarizes the Discussion again. Suggest returning to the more global issue of POCT uptake.

Response: Changes have been made to include commentary on the global POCT uptake (Lines 398-400).

Comments from Reviewer #2 

• Comment 1: Line 41-42: “We conducted 40 total interviews; 31 prior to POCT implementation; 9 post-implementation.”

This sentence could indicate that 40 different individuals were interviewed. Please make it more clear in the abstract that the 9 post-implementation interviews were with participants who had also provided pre-implementation interviews.

Response: This has been clarified (Lines: 42-43).

• Comment 2: Line 78-80: “Enquiries into facilitators and barriers to POCT adoption are particularly timely for sexual health: new rapid and POCTs for Chlamydia trachomatis (CT) and Neisseria gonorrhoeae (NG) are becoming more commonplace”

The authors reference the WHO diagnostic landscape for STIs report to justify this statement that new POCTs for chlamydia and gonorrhoea are “becoming more commonplace”. The term commonplace is a bit ambiguous here, and potentially implies that their adoption in clinical settings has become more frequent.

If the intent of this statement was to inform the reader that there are a number of POCTs for CT and NG currently available and more are likely in in the coming years, I suggest stating this more directly and potentially mentioning the pipeline of diagnostics specifically.

However, if the authors wished to state that the use of POCTs for CT/NG is in fact more commonplace in clinical practice, an alternative reference would be required to justify this.

Response: To reflect the reality, this sentence has been rewritten to suggest that POCTs are increasingly available for clinical adoption (Line 81).

• Comment 3. Line 102-110:

I appreciated the description of the reasons why services that initially expressed interest did not proceed. However, three of the six services that took part in the pre-implementation interviews “ultimately declined participation in the experimental model”, but the reasons for this are not discussed. For completion, please consider providing information on the reasons behind these three services not taking part, if available.

Response: Additional details have been added to specify the decision of the three services (Lines 113-117). 

• Comment 4: Line 127-128: “Interviews were conducted by AP, and analysis conducted by AP and SSF; both have >10 years’ experience with qualitative research.”

Given that this is a purely qualitative study, there is an onus to provide information on factors that may have influenced the analysis itself. Please provide more information on the background of researchers AP and SSF. The SRQR checklist recommends providing information on “researchers’ characteristics that may influence the research, including personal attributes, qualifications/experience, relationship with participants, assumptions, and/or presuppositions; potential or actual interaction between researchers’ characteristics and the research questions, approach, methods, results, and/or transferability”.

Response: More details about the interviewers have been added (Lines 200-207).

• Comment 5: Please provide a completed qualitative research checklist (SRQR or COREQ) as an appendix, and state and reference this in the methods.

Response: This has been attached as an appendix and referenced in the methods section (Lines 172-173). 

• Comment 6: I note that the names of the clinics have been withheld. If this is for confidentiality purposes, I suggest stating this explicitly in the methods.

Additionally, if the names are to be withheld, some background could still be provided on the clinics. For example, the cities/towns in which the six clinics are based could be stated without saying which location corresponds to which clinic. Alternatively, if they are all based in cities/large towns and if they are all based at large tertiary centres/smaller community clinics etc could be stated. This may help to provide additional context to the analysis.

Response: We have added a paragraph with description of the participating clinics (Lines 121-131).

• Comment 7: Line 146-150

The methods state that “A full list of themes is shown in Figure 2 and Figure 3.”

The titles of figures 2 and 3 refer to both “NVivo nodes” and “inductive themes”.

If the terms “NVivo nodes” and “themes” are equivalent, I suggest the use of a single term for consistency.

Response: This has been changed to ‘nodes’ (Lines 172-179).

• Comment 8: Line 153-154: “Invitation to participate was sent to professionals working in six sexual health services that showed interest in participating in the experimental model.”

The methods description of the interviewees is “professionals identified via the experimental model who self-identified as key in either the decision-making process for adoption of new technologies, and/or implementing new technologies, into their current service.”

These two descriptions in the methods and results sections are slightly incongruent. The methods section description suggests that all participants were key to adoption or implementation. However, the results section suggests that interest in the experimental model was the only pre-requisite.

I suggest re-writing one description or the other to ensure these descriptions are consistent and accurate.

Response: The results section was rewritten to match the earlier statement (Lines 182-184). 

• Comment 9: Line 301: “Our data indicate that a need for solutions to increase patient outcomes and overall wellbeing drove many HCPs’ desire for implementation of the binx POCT.”

I do not feel that this statement was adequately supported by the analysis presented in the results section and it seems to be the first mention of the drivers behind HCPs wanting to implement the binx POCT.

Please mention this in the results section if you wish for it to be a discussion point.

Response: This sentence was rewritten for clarity (Lines 348-351).

• Comment 10: The majority of respondents were healthcare professionals, and only one commissioner contributed to the pre-implementation data. I suggest commenting on how the breakdown of your respondents may have affected your results.

Additionally, did the authors feel that nine post-implementation interviewees was sufficient? Was data saturation reached?

Response: We have commented on the breakdown of our respondents (Lines: 385-390). 

With regards to data saturation, we believe that as we were only able to do follow-up interviews with those who participated in the clinical programme and there wasn’t enough uniformity in clinical use, reaching true thematic saturation was impossible and not applicable. We did however notice consistency in that the plans for implementation described in prior to implementation were largely followed as planned. 

Comment from Reviewer #3

• Comment: I don't see reference 14 anywhere in the text of the manuscript, but I would suggest not using a corporate press release. Similarly, reference 16 is from the manufacturer's package insert and thre are several peer-reviewed articles available describing the io so it would be better to use one of those.

Response: Reference 14 was removed from the list and reference 16 was replaced with a peer-reviewed article (Lines 325, and 471-472).

• Comment 2: How many io instruments were supplied to each clinic? If more than 1, is this part of the issue with space?

Response: All services received 1-2 readers. Although we had a respondent discuss the desire for more instruments but cite lack of space to accommodate this, we didn’t find this was a strong theme in interviews, as this wasn’t reported broadly, despite interviewer prompts. We cite our respondents concern around lack of space in the clinic in lines 324-326. 

• Comment 3: Did anyone mention patients having to wait >30 minutes because the instrument was already in use when their sample was taken? Is this an issue of concern for clinicians?

Response: More details on that issue have now been included (Lines 278-288). 

• Comment 4: Did anyone mention results integration concerns (or describe this as an advantage of the system)

Response: Respondents discussed that they manually entered patients’ results into their Electronic Patient Records (EPRs) and that this was a bit time consuming, but that this would not be part of routine integration of the POCT if it were adopted into the service. The io system is capable of integration with various EPR systems, but time and paperwork for getting permissions and setting up that integration wasn’t feasible for a short trial implementation period. Consequently, we don’t have any data on this in real practice, and as such it is not included in our reporting.

We look forward to hearing from you in due time regarding our submission and to respond to any further questions and comments you may have. 

Sincerely,

Dr Agata Pacho (on behalf of all authors)

---

## [Decision Letter · Decision Letter 1]

28 Feb 2022

Facilitators and barriers for clinical implementation of a 30-minute point-of-care test for Neisseria gonorrhoeae and Chlamydia trachomatis into clinical care: a qualitative study within sexual health services in England

PONE-D-21-14685R1

Dear Dr. Sebastian,

We’re pleased to inform you that your manuscript has been judged scientifically suitable for publication and will be formally accepted for publication once it meets all outstanding technical requirements.

Kind regards,

Carlos Miguel Rios-González, Ph.D

Academic Editor

PLOS ONE

Additional Editor Comments (optional):

Dear Author

Thank you very much for submitting your manuscript to our journal, I am forwarding your article with the comments of the reviewers.

Kind regards.

Reviewers' comments:

Reviewer's Responses to Questions

**Comments to the Author**

1. If the authors have adequately addressed your comments raised in a previous round of review and you feel that this manuscript is now acceptable for publication, you may indicate that here to bypass the “Comments to the Author” section, enter your conflict of interest statement in the “Confidential to Editor” section, and submit your "Accept" recommendation.

Reviewer #2: All comments have been addressed

Reviewer #3: All comments have been addressed

2. Is the manuscript technically sound, and do the data support the conclusions?

Reviewer #2: Yes

Reviewer #3: (No Response)

3. Has the statistical analysis been performed appropriately and rigorously? 

Reviewer #2: N/A

Reviewer #3: (No Response)

4. Have the authors made all data underlying the findings in their manuscript fully available?

Reviewer #2: Yes

Reviewer #3: (No Response)

5. Is the manuscript presented in an intelligible fashion and written in standard English?

Reviewer #2: Yes

Reviewer #3: (No Response)

6. Review Comments to the Author

Reviewer #2: (No Response)

Reviewer #3: (No Response)

7. PLOS authors have the option to publish the peer review history of their article (what does this mean?). If published, this will include your full peer review and any attached files.

Reviewer #2: No

Reviewer #3: No

---

## [Editor Report · Acceptance letter]

2 Mar 2022

PONE-D-21-14685R1 

Facilitators and barriers for clinical implementation of a 30-minute point-of-care test for Neisseria gonorrhoeae and Chlamydia trachomatis into clinical care: a qualitative study within sexual health services in England 

Dear Dr. Fuller:

I'm pleased to inform you that your manuscript has been deemed suitable for publication in PLOS ONE. Congratulations! Your manuscript is now with our production department. 

Kind regards, 

on behalf of

Dr. Carlos Miguel Rios-González 

Academic Editor

PLOS ONE